# FITS: Conditional Diffusion Model for Irregular Time Series Forecasting with Pseudo-future Exogenous Covariates

## Abstract

Irregular multivariate time series (IMTS) present unique challenges due to non-uniform intervals and different sampling rates. While existing methods struggle to capture both long-term dynamics and cross-channel dependencies under such irregularities, we tackle this by formulating time series forecasting as a conditional generation problem and introducing FITS, a conditional diffusion model for IMTS forecasting that leverages pseudo-future exogenous covariates. Our approach incorporates two key innovations. First, we propose a novel entropy-aware adaptive patching scheme that generates data-driven segments with dynamic boundaries determined by the information density. This scheme overcomes the limitations of traditional fixed-length or fixed-span segmentation in preserving continuous local semantics and modeling inter-time series correlations. Second, we develop a transformer-based prior knowledge extractor that captures forward-looking covariate dependencies via a novel cross-variate attention mechanism. The transformer structure is integrated into the conditional diffusion generative process as a unified framework, enabling precise distributional forecasting for IMTS. Extensive experiments on multiple datasets with four evaluation metrics validate the effectiveness of FITS.

## 1 Introduction

Time series forecasting (TSF) plays a crucial role in numerous real-world applications, facilitating data-driven decision-making across diverse fields. It is widely utilized in domains such as stock price prediction (Li et al., 2024a), weather prediction, transportation planning (Guo et al., 2022), and healthcare. Many approaches, such as autoregressive models (Salinas et al., 2020) and sequence-to-sequence modeling (Wen et al., 2017), frame forecasting as a conditional generative task. In particular, diffusion-based generative models have attracted considerable attention owing to their capabilities in image, video, and text generation (Ho et al., 2020a; Dhariwal & Nichol, 2021; Kong et al., 2021).

Most existing time series diffusion models are designed for regularly sampled time series, such as Li et al. (2024c); Shen et al. (2024); Wang et al. (2025), however, when dealing with sparse and irregularly observed data, there are several obstacles: (1) how to capture irregularities in intra-series dependencies and asynchronies in inter-series correlations amid varying time intervals between adjacent observations; (2) how to extract critical insights from all available historical data, which can then serve as prior knowledge to capture covariate dependencies in both forward and reverse processes within the diffusion model. While prior studies such as Li et al. (2024b) and Shen & Kwok (2023) have proposed effective conditional embeddings to guide the diffusion process, when the conditional inputs (e.g., historical observations) are highly sparse, models face challenges in extracting adequate contextual information as they are unable to capture the temporal dependencies, compromising the reliability of time series prediction.

To this end, we propose a conditional diffusion model for **i**rregular **t**ime **s**eries forecasting with pseudo-future exogenous covariates (FITS), which integrates a transformer-enhanced modeling approach to capture the forward - backward covariate dynamics. It then leverages this model to generate pseudo forecasts of the target variable, which essentially serve as conditional guidance for generating the unobserved segments of sparse time series, supporting downstream prediction tasks (Fig. 1).

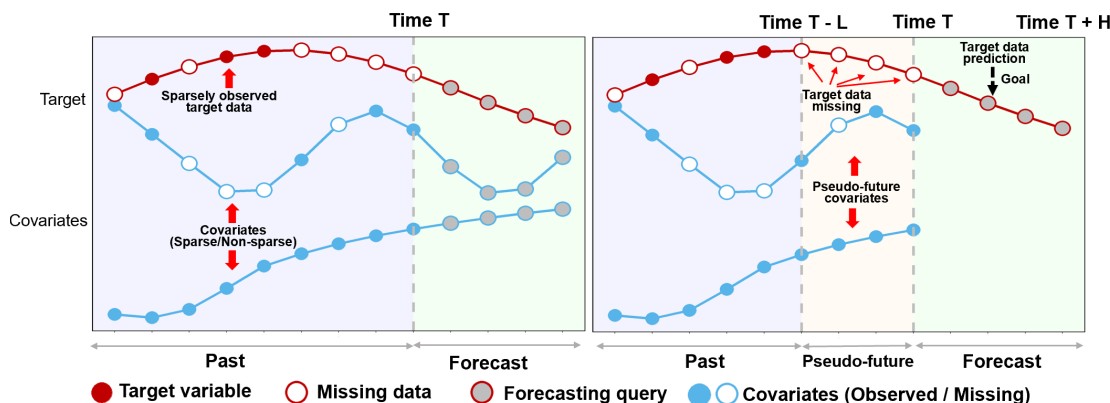

Figure 1: The comparison of regular multivariate time series forecasting and IMTS forecasting frmework considered in this work. (a) The goal of regular multivariate time series forecasting is to simultaneously forecast all the variables in the system. (b) The goal of this work is to forecast the sparse and irregularly observed target time series, given all the external covariates, and the covariates could be either sparse or non-sparse.

Our contributions are summarized as follows: We introduce FITS, a conditional diffusion model designed for forecasting sparse and irregularly observed time series. Specifically: (1) FITS incorporates an adaptive entropy-based patching approach tailored to irregular time series, which leverages local semantic granularities and enables more accurate modeling of inter-series correlations. (2) FITS also employs a transformer-based predictive model learned from external covariates, equipped with forward-looking cross-variate attention mechanisms. During the reverse diffusion process, this learned model is leveraged as a conditional representation to generate accurate probability distributions of future time series. (3) In our experiments, besides standard evaluation metrics such as mean squared error (MSE) and mean absolute error (MAE), we employed Prediction Interval Coverage Probability (PICP) (Yao et al., 2019) and Quantile Interval Coverage Error (QICE) (Han et al., 2022) as metrics for the probabilistic multivariate time series forecasting task. Extensive experiments demonstrate that FITS outperforms state-of-the-art time series diffusion models and performs better than or comparable to various advanced time series prediction models.

## 2 RELATED WORK

### 2.1 IRREGULAR MULTIVARIATE TIME SERIES FORECASTING

Existing works have primarily focused on IMTS classification (Yalavarthi et al., 2022; Horn et al., 2020; Tashiro et al., 2021), imputation (Shukla & Marlin, 2021a; Yalavarthi et al., 2023) and forecasting (Zhang et al., 2023; Mercatali et al., 2024; Yalavarthi et al., 2024). To summarize the core mechanism of the IMTS forecasting methods in addressing the data irregularities, some authors proposed novel data preprocessing and representation methods, for example, in the patching-based approach, the input time series is represented as matrices with temporal and variable dimensions, and model components are designed to learn

dependencies along both dimensions Zhang et al. (2024), however, in the case of sparsely observed time series, the number of observations within a patch may be scarce, resulting in an excessive number of uninformative patches under a temporal resolution; there are also other non-patching approaches that use bipartite graphs (Yalavarthi et al., 2024), or hypergraphs (Li et al., 2025), but their model architectures restrict the ability to capture dependencies in high dimensional or highly sparse IMTS.

In addition to data representation methods for IMTS forecasting, some authors also proposed novel deep architectures and attention mechanisms. For example, *T-PATCHGNN* (Zhang et al., 2024) proposed a time-adaptive graph neural network to model the dynamic intra-patch and inter-patch dependencies. *Warpformer* (Zhang et al., 2023) proposed a doubly self-attention module within the transformer framework for representation learning on multiple sampling granularities. *ContiFormer* (Chen et al., 2023) adopted continuous-time Neural ordinary differential equations (ODEs) within the attention mechanism of Transformers to capture the temporal dynamics of the underlying IMTS system. These methods often presume a specific form of dependency, which introduces significant restrictiveness and fails to accommodate considerations of complex hierarchical, higher-order or multi-scale dependencies.

## 2.2 TIME SERIES DIFFUSION MODELS

The Denoising Diffusion Probabilistic Models proposed by Ho et al. (2020b) has become a powerful tool for time series modeling (Lin et al., 2024), due to their advantages in fine-grained temporal modeling. Many recent time-series diffusion models have focused on designing effective conditional embeddings to guide the reverse process (Li et al., 2024c; Tashiro et al., 2021; Rasul et al., 2021). For example, TimeGrad (Rasul et al., 2021) employs the hidden state from an RNN as the conditional embedding, Li et al. (2024c) utilized vanilla transformers to extract a representation from historical data, which is then used as a prior knowledge to recover the full distribution of future time series. In addition, Shen & Kwok (2023) further incorporated parts of the ground-truth future predictions for conditioning, which introduces additional inductive bias in the conditioning module for more accurate time series prediction. Shen et al. (2024) also considered other unique time series properties and proposed a multi-resolution diffusion model corresponding to a sequence of fine-to-coarse trend.

So far, the existing works on time series diffusion models have been focused on regularly sampled time series data, in the context of IMTS, representations extracted from historical data may fail to capture the underlying trends of the sequence, leading to a lack of reliable prior guidance, making it prone to generating sequences that are disconnected from historical patterns. Furthermore, in terms of model training during the reverse process, it is difficult to generate the desired series when there are limited fine granularity information (Coletta et al., 2023), which may provide unreliable underlying inputs for the multi-resolution framework and thus undermining the consistency of the overall trend.

## 3 PROPOSED METHOD

In this work, we assume that the total length of the observed time series is $T$, where the historical observed target time series $\mathbf{x}_{0:T-L}$ ($0 < L < T$) is sparse and irregularly sampled, with its last valid observation recorded at time $T - L$. Furthermore, we consider multiple exogenous covariates $\mathbf{z}_{0:T} \in \mathbb{R}^{T \times C}$, where $C$ represents the dimensionality of the exogenous covariates; by definition, any time series that provides predictive value for the prediction target is classified as an exogenous covariate. The proposed diffusion-based forecasting framework aims to predict the future segment $\mathbf{x}_{T:T+H}$ using a model $\mathcal{F}_\theta$ that specifically captures all available information embedded in the historical observed time series $\mathbf{x}_{0:T-L}$ and exogenous covariates $\mathbf{z}_{0:T}$.

$$\widehat{\mathbf{x}}_{T:T+H} = \mathcal{F}_\theta \left( \mathbf{x}_{0:T-L}, \mathbf{z}_{0:T} \right). \tag{1}$$

Fig. 2 shows an overview of the proposed model.

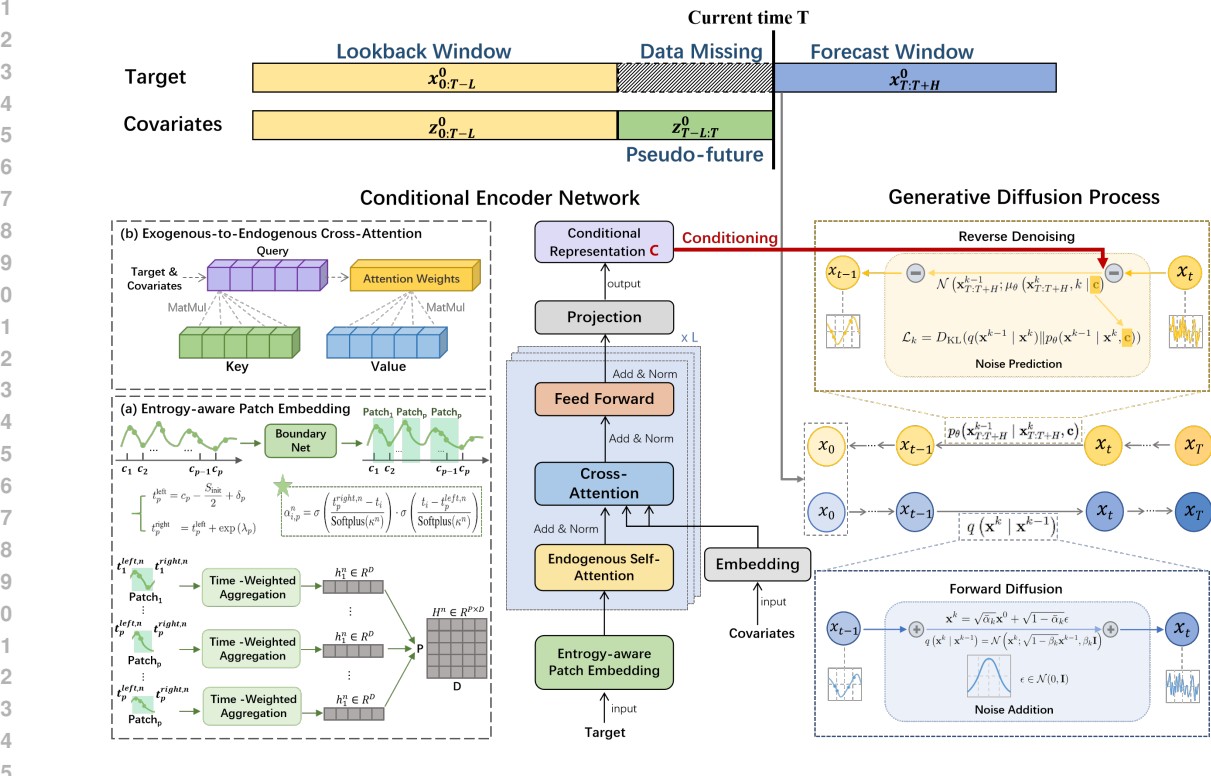

Figure 2: Overall framework of the proposed FITS framework.

## 3.1 FORWARD DIFFUSION PROCESS

During model training, the objective of the forward diffusion is to diffuse the "future" time steps $\mathbf{x}_{T:T+H}$ of the target time series. At the $k$-th step of the forward process, $\mathbf{x}^k$ is parameterized by adding noise to the previous diffusion step $k-1$, scaled by $\sqrt{1-\beta_k}$:

$$q\left(\mathbf{x}^k \mid \mathbf{x}^{k-1}\right) = \mathcal{N}\left(\mathbf{x}^k; \sqrt{1-\beta_k}\mathbf{x}^{k-1}, \beta_k\mathbf{I}\right), \quad k = 1, \ldots, K, \tag{2}$$

with $\beta_t \in (0, 1)$ representing the noise variance following a predefined schedule. It can be shown that:

$$q\left(\mathbf{x}^k \mid \mathbf{x}^0\right) = \mathcal{N}\left(\mathbf{x}^k; \sqrt{\bar{\alpha}_k}\mathbf{x}^0, (1-\bar{\alpha}_k)\mathbf{I}\right), \tag{3}$$

where $\bar{\alpha}_k = \Pi_{s=1}^k \alpha_s$, and $\alpha_k = 1 - \beta_k$. Then, $\mathbf{x}^k$ is given as:

$$\mathbf{x}^k = \sqrt{\bar{\alpha}_k}\mathbf{x}^0 + \sqrt{1-\bar{\alpha}_k}\epsilon, \quad \epsilon \in \mathcal{N}(0, \mathbf{I}). \tag{4}$$

The subscript of $\mathbf{x}_{T:T+H}$ is omitted for notational simplicity.

## 3.2 CONDITIONING THE BACKWARD DENOISING PROCESS

Existing time series diffusion models typically incorporate either the original historical observation segment $\mathbf{x}_{0:T}$ (Tashiro et al., 2021) or a derived representation $\mathcal{F}(\cdot)$ from historical data (Li et al., 2024b) as

input to their conditioning networks. In contrast, this study proposes to leverage the evolutionary dynamics embedded in external covariates, which capture the relationship from $\mathbf{z}_{0:T-L}$ to $\mathbf{z}_{T-L:T}$. This latent process characterize the potential variation patterns of the target variable from the historically observed part $\mathbf{x}_{0:T-L}$ to the "pseudo-future" segment $\mathbf{x}_{T-L:T}$, thereby facilitating predictive inference.

### 3.2.1 ENTROPY-AWARE PATCHING AND ENCODING FOR IRREGULAR TIME SERIES

In this subsection, we propose a novel information density-based patching and encoding approach applied to all variables. For IMTS, it is difficult to capture the local dynamic granular scemantics due to discretionary segmentation of continuous observations, which hinders the effective extraction of low-dimensional latent factors and state evolution patterns. For example, a patient's sudden health deterioration may be segmented across two time windows, which fragments this critical pattern and prevents it from being fully captured.

**Entropy-aware module to compute dynamic window boundaries.** To fully leverage temporal information, we first enrich each raw observation by filling the missing points with zero. Motivated by Liu et al. (2025), assume the historical observation is initially divided into $P$ patches with length $S_{\text{init}} = T/P$. For each patch $p$, the initial reference center is $c_p = (p - 0.5) \cdot (T/P)$, and the window boundaries can be computed as:

$$t_p^{\text{left}} = c_p - \frac{S_{\text{init}}}{2} + \delta_p, \quad t_p^{\text{right}} = t_p^{\text{left}} + \exp(\lambda_p). \tag{5}$$

In this work, we propose a novel boundary network (BoundaryNet) based on a sample entropy (SampEn) measure to specifically learn the parameters $\delta_p$ and $\lambda_p$ in Eq. (5). Specifically, the SampEn measure proposed by Richman & Moorman (2000) quantifies the information richness of a time series: a higher entropy value indicates a more complex series that harbors dense implicit information. According to the definition of SampEn, we first compute the number of matching subsequences for a given embedding dimension $m$ and similarity tolerance $r$, denoted as *Mat*. For the target sequence, we then utilize a lightweight MLP network to map *Mat* to the latent space, enabling the calculation of the two scalar boundary parameters, given below:

$$[\delta_p, \lambda_p] = \text{Linear}_{\text{output}}\left(\text{SiLU}\left(\text{Linear}_{\text{hidden}}\left(Mat\right)\right)\right). \tag{6}$$

Substitute Eq. (6) into Eq. (5), we can effectively compute the dynamically adjusted window boundaries based on the information density.

**Adaptive patch representations.** After defining the dynamic temporal windows, using the method proposed by Liu et al. (2025), we calculate a relevance weight $\alpha_{i,p}$ using $[\delta_p, \lambda_p]$ for each observation $i$ in patch $p$ and arrive at the final representation:

$$\bar{h}_p = \frac{\sum_{i=1}^{L_p} \alpha_{i,p} \cdot \tilde{v}_i}{\sum_{i=1}^{L} \alpha_{i,p} + \epsilon} \in \mathbb{R}^{1+D_{te}}, \tag{7}$$

where $L_p$ denotes the number of observations in patch $p$, and $\tilde{v}_i = \text{Concat}(\mathbf{x}_p(t_i), \text{TE}(t_i))$, $\text{TE}(t_i) \in \mathbb{R}^{D_{te}}$ denotes the learnable time embedding. Then, $\bar{h}_p$ is projected into the model's uniform hidden space via a linear layer: $h_p = \text{Linear}_D(\bar{h}_p) \in \mathbb{R}^D$. Therefore, we have for the whole sequence: $H = [h_1, \ldots, h_P] \in \mathbb{R}^{P \times D}$.

### 3.2.2 LEARNING CONDITIONAL REPRESENTATION THROUGH RECONCILIATING TARGET AND EXOGENOUS INFORMATION

In this work, a transformer is utilized as a prior knowledge extractor, capturing covariate-dependence in the reverse process within the diffusion model. In addition to the patch representation $H$ derived in the previous section, the entire target time series $\mathbf{x}_{0:T}$ is also embedded into one single series-level global token embedding $\mathbf{G}_{\text{tar}}$ via the same trainable linear MLP projector.

**Intra-series self-attention.** In the patch-level attention, we apply multi-head attention with causal masking to all variables to capture their intra-variate cross-time dependency. Taking the target variable as an example, and dropping layer index for brevity, this can be formalized as:

$$\widetilde{\mathbf{H}}_{:}^{\text{pat}} = \text{LN}\left(\mathbf{H}_{:} + \text{MHA}\left(\mathbf{H}_{:}, \mathbf{H}_{:}, \mathbf{H}_{:}\right)\right),$$
$$\mathbf{H}_{:} = \text{LN}\left(\widetilde{\mathbf{H}}_{:}^{\text{pat}} + \text{FFN}\left(\widetilde{\mathbf{H}}_{:}^{\text{pat}}\right)\right), \tag{8}$$

where $\mathbf{H}_{:}$ denotes the collective token embeddings of a variable at all patch steps, LN denotes layer normalization, MHA$(\mathbf{Q}, \mathbf{K}, \mathbf{V})$ denotes the multi-head attention layer where $\mathbf{Q}$, $\mathbf{K}$, and $\mathbf{V}$ serve as queries, keys and values, and FFN denotes a feed-forward network. In addition, we also employ a series-level global token embedding $\mathbf{G}_{\text{tar}}$, which serves as a bridge that connects the patches in the target variable and the exogenous variables (Wang et al., 2024). Accordingly, we also employ a variate-to-patch attention $\mathbf{H}_{:}^{\text{var-to-pat}}$ and a patch-to-variate attention $\mathbf{G}^{\text{pat-to-var}}$, which offers a holistic perspective of the temporal dependencies inherent to the target variable, while also enabling enhanced interactions with exogenous variables that exhibit arbitrary irregularity. **Inter-series cross-attention.** Assume the last observed data point of the target variable occurs at time $T - L$; $\mathbf{z}_{T-L:T}$ thus constitutes a relative future segment relative to $\mathbf{x}_{0:T-L}$. To this end, we redesign the cross-attention layer: the global token of the target variable, $\mathbf{G}_{\text{tar}}$, remains the query (Q), while exogenous variables are split into two segments for the key (K) and value (V), where the embedding of the historical segment $\mathbf{z}_{0:T-L}$ serves as K and the embedding of the pseudo-future segment $\mathbf{z}_{L:T}$ serves as V. The learned global token of the target acts as a bridge to integrate and filter exogenous information, ensuring that only relevant insights support the prediction of the target variable.

### 3.2.3 CONDITIONING NETWORK

Following Shen & Kwok (2023), using the transformer network $\mathcal{T}(\cdot)$ derived from the previous sections, we adopt the future mixup strategy which combines the past information's mapping $\hat{\mathbf{x}}_{T:T+H} = \mathcal{T}(\mathbf{x}_{0:T-L})$ with the future ground-truth $\mathbf{x}_{T:T+H}^0$, which is only available during training. At diffusion step $k$, it produces the conditioning signal $\mathbf{c}$ as:

$$\mathbf{c} = \mathbf{m}^k \mathcal{T}\left(\mathbf{x}_{0:T-L}\right) + \left(1 - \mathbf{m}^k\right) \mathbf{x}_{T:T+H}^0. \tag{9}$$

Here, $\mathbf{m}^k \in [0, 1)^{1 \times H}$ is a mixing coefficient randomly sampled from the uniform distribution on $[0, 1)$. During inference, $\mathbf{x}_{T:T+H}^0$ is no longer available, and the condition $\mathbf{c}$ is set to $\mathcal{T}(\mathbf{x}_{0:T-L})$.

### 3.3 DENOISING REVERSE PROCESS

The reverse denoising process is a markov chain. At the $k$-th denoising step, $\mathbf{x}_{T:T+H}^{k-1}$ is generated from $\mathbf{x}_{T:T+H}^k$ by sampling from the following normal distribution, subject to the conditional representation $\mathbf{c}$:

$$p_\theta\left(\mathbf{x}_{T:T+H}^{k-1} \mid \mathbf{x}_{T:T+H}^k, \mathbf{c}\right) = \mathcal{N}\left(\mathbf{x}_{T:T+H}^{k-1}; \mu_\theta\left(\mathbf{x}_{T:T+H}^k, k \mid \mathbf{c}\right), \Sigma_\theta\left(\mathbf{x}_{T:T+H}^k, k\right)\right), \tag{10}$$

where the variance $\Sigma_\theta\left(\mathbf{x}_{T:T+H}^k, k\right)$ is fixed to $\sigma_k^2 \mathbf{I}$. The goal of this reverse process is to learn this mean function $\mu_\theta(\mathbf{x}_{T:T+H}^k, k)$, which effectively produces $\mathbf{x}_{T:T+H}^{k-1}$ close to the ground truth. Through iterative denoising steps, the prediction result $\hat{x}_{T:T+H}^0$ is ultimately recovered to match the distribution of the original time series. To train the diffusion model, considering Eqs. (2) and (10), one uniformly samples $k$ from $\{1, 2, \ldots, K\}$ and then minimizes the KL (Kullback-Leibler) divergence:

$$\mathcal{L}_k = D_{\text{KL}}(q(\mathbf{x}^{k-1} \mid \mathbf{x}^k) \| p_\theta(\mathbf{x}^{k-1} \mid \mathbf{x}^k, \mathbf{c})), \tag{11}$$

where $q(\mathbf{x}^{k-1} \mid \mathbf{x}^k)$ is the ground-truth conditional data distribution.

Then, the training objective in (11) is then formulated as:

$$\mathcal{L}_k = \frac{1}{2\sigma_k^2} \left\| \tilde{\mu}_k \left( \mathbf{x}^k, \mathbf{x}^0, k \right) - \mu_\theta \left( \mathbf{x}^k, k \mid \mathbf{c} \right) \right\|^2. \tag{12}$$

The estimation of $\mu_\theta(\mathbf{x}^k, k \mid \mathbf{c})$ can be computed via a noise prediction model $\epsilon_\theta(\mathbf{x}^k, k)$ following Benny & Wolf (2022).

During inference, a noise vector $\mathbf{x}_{1:H}^K \sim \mathcal{N}(\mathbf{0}, \mathbf{I})$ is generated, and through the reverse denoising process, we can obtain the final prediction result $\hat{\mathbf{x}}_{T:T+H}^0$.

## 4 EXPERIMENTS

In this section, we perform extensive experiments to compare the proposed FITS with recent 6 state-of-the-art (SOTA) time series prediction models on 7 commonly used real-world datasets.

### 4.1 SETUP

**Benchmark datasets.** Experiments were performed on 7 public benchmark datasets with different levels of multivariate correlations. The datasets include: (i) **Electricity Price Forecasting Dataset (EPF)** (5 sub-datasets from different major power markets) (Lago et al., 2021). (ii) **Exchange** (Daily exchange rates of eight different countries) (Lai et al., 2018). (iii) **Weather** (21 meteorological variables from Germany) (Zhou et al., 2021). Due to space constraints, detailed descriptions of the datasets are deferred to Appendix B.1. Appendix B.1 also includes the process of downsampling the original data to arrive at the sparse and irregular time series used in this work, which in turn contextualizes the data preparation aligned with our research focus. The data are processed using two random missingness strategie.

**Baselines.** To establish a comprehensive benchmark for our proposed FITS method, we select baselines from four methodological domains. Specifically, we include: (i) **Time series diffusion models:** CSDI (Tashiro et al., 2021); Transformer-Modulated Diffusion Model (TMDM) (Li et al., 2024b); Diffusion-TS (Yuan & Qiao, 2024). (ii) **Time series transformers:** Crossformer (Zhang & Yan, 2023). (iii) **Other time series forecasting methods:** TiDE (Das et al., 2023); DLinear (Zeng et al., 2023). mTAN (Shukla & Marlin, 2021b). See Appendix B.2 for more details about the baselines.

**Implementation details.** In our experiments, we employed a linear noise schedule with $\beta_1 = 10^{-4}$ and $\beta_K = 0.02$, setting the number of diffusion timesteps to $K = 1000$. We approximated the data distribution using 100 samples, and all experiments were repeated 5 times with seeds $\{1, 2, 3, 4, 5\}$. The model was trained using the Adam optimizer with a learning rate of $10^{-4}$ and a batch size of 64. Additional details are given in Appendix B.3.

### 4.2 MAIN RESULTS

#### 4.2.1 PROBABILISTIC FORECASTING

To intuitively illustrate the probabilistic distribution forecasting capabilities of the models, we present the forecasting results of our proposed FITS model alongside three comparative baseline models in Figure 4. Specifically, we visualize the 50% and 90% prediction intervals (denoted by dark green and light green, respectively) and overlay the true observed values for direct reference. It is worth noting that certain baseline models were originally devised for generative tasks rather than dedicated probabilistic forecasting; however, their authors have asserted that these models are capable of yielding probabilistic forecasting results (Yuan & Qiao, 2024).

As evidenced by the visualization, FITS demonstrates superior performance in probabilistic distribution forecasting. This advantage can be attributed to the design of our conditional estimation module, which enables more accurate mean estimation even when the input data suffers from temporal misalignment and high proportions of missing values. In particular, the inter-series cross-attention component embedded within this module facilitates the model's effective extraction and utilization of latent information in pseudo-future data, thereby enhancing forecasting reliability. Nevertheless, in scenarios where there exist unobserved gaps between the historical information window and the target forecasting window, all models encounter heightened challenges in capturing future trend dynamics, resulting in elevated predictive uncertainty.

To quantitatively analyze the models' probabilistic forecasting capabilities, we adopted CRPS (Continuous Ranked Probability Score) and QICE (Quantile Interval Coverage Error) as evaluation metrics, following the approach of Li et al. (2024b). For both metrics, smaller values indicate better performance. Table 1 shows the CPRS and QICE on the time series. Notably, our model achieves the optimal performance on nearly all datasets, with its CRPS and QICE values consistently remaining at the lowest level among all compared models, fully demonstrating its superior probabilistic forecasting capability.

Table 1: Performance comparisons in terms of QICE and CRPS. The best results are boldfaced, and the suboptimal results are underlined. The table presents both the scenarios of no missing values and random missingness of 0.5.

| | | Weather | | Exchange | | NP | | PJM | | BE | | FR | | DE | |
|---|---|---|---|---|---|---|---|---|---|---|---|---|---|---|---|
| | | QICE | CRPS | QICE | CRPS | QICE | CRPS | QICE | CRPS | QICE | CRPS | QICE | CRPS | QICE | CRPS |
| CSDI | No-Missing | 13.91 | 0.735 | 10.91 | **0.178** | 1.96 | **0.279** | 16.75 | 0.603 | 14.63 | 0.393 | 15.60 | 0.377 | 13.34 | 0.683 |
| | RM=0.5 | 14.90 | 0.763 | 10.76 | **0.189** | 2.394 | 0.324 | 14.37 | 0.624 | 14.33 | 0.452 | 15.32 | 0.386 | 13.56 | 0.769 |
| TMDM | No-Missing | 10.86 | 0.485 | 7.439 | 0.516 | 5.609 | 0.593 | 4.541 | 0.209 | 6.096 | 0.353 | 5.125 | 0.294 | 4.577 | 0.407 |
| | RM=0.5 | 12.20 | 0.554 | 7.305 | 0.488 | 5.453 | 0.503 | **2.511** | 0.178 | 9.001 | 0.532 | 8.795 | 0.388 | 4.488 | 0.410 |
| Diffusion-TS | No-Missing | 12.13 | 0.532 | 15.90 | 1.310 | 9.692 | 0.612 | 15.36 | 0.218 | 8.236 | 0.401 | 9.486 | 0.376 | 13.96 | 0.785 |
| | RM=0.5 | 13.690 | 0.543 | 15.27 | 1.040 | 10.25 | 0.635 | 15.20 | 0.256 | 8.569 | 0.490 | 10.50 | 0.358 | 12.30 | 0.813 |
| FITS | No-Missing | **3.275** | **0.409** | **4.966** | 0.354 | **1.800** | 0.287 | **3.007** | **0.176** | **2.854** | **0.226** | **3.739** | **0.182** | **2.162** | **0.390** |
| | RM=0.5 | **4.170** | **0.497** | **4.979** | 0.359 | **1.924** | 0.277 | 2.976 | **0.177** | **3.023** | **0.226** | **3.830** | **0.190** | **1.024** | **0.377** |

### 4.2.2 NON-PROBABILISTIC FORECASTING

Tables 2 and 3 present the Mean Squared Error (MSE) and the Mean Absolute Error (MAE) results. It can be observed that the performance improvement of the model is particularly significant on more complex datasets such as BE and FR. Overall, the FITS model ranks the highest compared to the baselines under two missingness rates. It should be noted that the model does not achieve performance improvement on long-term forecasting datasets such as Exchange rate, and this may be because there are no complex inter-dependencies between variables in such datasets, leading to the introduction of noise by the inter-variable attention mechanism in the conditional estimation model. TiDE and DLinear, the two channel-independent models, achieved the optimal and suboptimal performance respectively, which also corroborates this point. In contrast, the covariates of the EPF dataset have been confirmed to indeed have a positive effect on target prediction, so our model has achieved better performance on this dataset. Furthermore, we also found that all diffusion models in the baselines perform poorly, which indicates that the current diffusion models are weaker than general models in terms of mean prediction ability.

### 4.2.3 MODEL EFFICIENCY

To systematically evaluate how model efficiency varies with the number of variables, we conducted predictive performance comparisons against the baseline models using two distinct datasets: EPF-DE (comprising 3 variables) and Weather (consisting of 21 variables), shown in Tables 4 and 5.

Table 2: Performance comparisons in terms of MAE and MSE. The best results are boldfaced, and the suboptimal results are underlined. The table presents the scenarios of RM=0.3 for both target and covariates.

| | PJM | | BE | | DE | | FR | | NP | | Eeather | | Exchange | | Ranking |
| | MSE | MAE | MSE | MAE | MSE | MAE | MSE | MAE | MSE | MAE | MSE | MAE | MSE | MAE | |
|---|---|---|---|---|---|---|---|---|---|---|---|---|---|---|---|
| Tide | 0.183 | 0.286 | 0.633 | 0.393 | 0.840 | 0.591 | 0.522 | _0.335_ | 0.532 | _0.482_ | 0.928 | 0.683 | 0.344 | _0.456_ | 4.14 |
| Dlinear | 0.192 | 0.295 | 0.637 | 0.413 | 0.895 | 0.612 | 0.547 | 0.359 | 0.560 | 0.503 | 0.927 | 0.679 | _0.341_ | 0.461 | 5.36 |
| Crossformer | 0.185 | 0.256 | _0.512_ | _0.359_ | **0.692** | **0.520** | _0.486_ | **0.314** | 0.523 | 0.492 | **0.574** | **0.552** | 0.677 | 0.504 | 2.94 |
| mTAN | 0.219 | 0.277 | 0.700 | 0.545 | 0.955 | 0.963 | 0.827 | 0.589 | 0.918 | 0.762 | 1.009 | 0.701 | 1.218 | 0.972 | 7.64 |
| CSDI | 0.221 | 0.322 | 0.540 | 0.447 | 0.944 | 0.652 | 0.576 | 0.356 | **0.515** | **0.469** | 2.136 | 1.361 | 4.245 | 1.609 | 6.21 |
| TMDM | **0.151** | 0.257 | 0.640 | 0.460 | 2.094 | 0.966 | 0.553 | 0.368 | 1.101 | 0.874 | 0.820 | 0.611 | 0.849 | 0.786 | 5.79 |
| Diffusion-ts | 0.169 | 0.276 | 0.651 | 0.480 | 0.806 | 0.588 | 0.678 | 0.460 | 1.252 | 0.853 | 0.768 | 0.658 | 1.791 | 1.223 | 6.21 |
| FITS | _0.156_ | **0.250** | **0.510** | **0.358** | _0.723_ | _0.560_ | **0.469** | 0.357 | 0.529 | 0.513 | 0.813 | 0.596 | **0.253** | **0.414** | **2.29** |

Table 3: Performance comparisons in terms of MAE and MSE. The best results are boldfaced, and the suboptimal results are underlined. The table presents the scenarios of RM=0.5 for both target and covariates.

| | PJM | | BE | | DE | | FR | | NP | | Weather | | Exchange | | Ranking |
| | MSE | MAE | MSE | MAE | MSE | MAE | MSE | MAE | MSE | MAE | MSE | MAE | MSE | MAE | |
|---|---|---|---|---|---|---|---|---|---|---|---|---|---|---|---|
| Tide | 0.192 | 0.294 | _0.617_ | _0.412_ | 0.862 | 0.598 | 0.666 | 0.387 | 0.638 | _0.525_ | 0.963 | 0.696 | _0.402_ | _0.491_ | 4.29 |
| Dlinear | 0.199 | 0.304 | 0.650 | 0.428 | 0.953 | 0.631 | 0.591 | 0.364 | 0.671 | 0.545 | 0.995 | 0.701 | 0.451 | 0.526 | 5.86 |
| Crossformer | 0.238 | 0.267 | 0.638 | **0.374** | **0.716** | **0.539** | 0.511 | 0.309 | **0.544** | **0.500** | **0.623** | **0.567** | 0.768 | 0.713 | 4.00 |
| mTAN | 0.241 | 0.264 | 0.663 | 0.435 | 1.082 | 0.712 | 0.810 | 0.602 | 1.304 | 0.933 | 0.938 | 0.725 | 1.169 | 0.967 | 7.43 |
| CSDI | 0.220 | 0.324 | 0.675 | 0.420 | 0.981 | 0.667 | _0.466_ | 0.372 | **0.588** | 0.526 | 4.153 | 2.336 | 2.769 | 1.325 | 6.29 |
| TMDM | 0.271 | 0.362 | 0.987 | 0.592 | 2.089 | 0.846 | 0.721 | 0.497 | 1.806 | 1.035 | 0.820 | 0.611 | 0.931 | 0.873 | 7.86 |
| Diffusion-ts | 0.194 | 0.291 | 0.867 | 0.646 | 0.851 | 0.583 | 1.546 | 0.823 | 1.375 | 0.900 | _0.757_ | _0.603_ | 1.522 | 1.123 | 6.29 |
| FITS | **0.174** | **0.254** | **0.590** | 0.460 | _0.837_ | _0.597_ | **0.469** | **0.357** | 0.7353 | 0.595 | 1.156 | 0.761 | **0.269** | **0.412** | **3.43** |

Several observations can be drawn from the two tables. TiDE, as an MLP-based model, is inherently a lightweight architecture, characterized by relatively short training and inference times as well as modest memory requirements. Crossformer adopts a transformer-based architecture with attention mechanisms, and the cross-attention calculations involved incur substantial memory overhead.

For all diffusion-based benchmark methods, their forward-reverse diffusion process operates on the entire time series, which results in these models having the highest time and memory consumption. In contrast, our proposed method leverages a conditional denoising design that restricts diffusion computations to the forecasting series exclusively, thereby improving time efficiency relative to its closest diffusion-based counterparts. Moreover, our novel attention mechanism structure enables a substantial reduction in memory usage when compared to the TMDM method, which also leverages the forecasting series as the diffusion target.

### 4.2.4 ABLATION STUDY

We compared prediction results of one full model and three ablation variants in Table 6. The **rp-atten** variant replaces the proposed inter-variable attention with standard cross-attention, leading to performance degradation; **w/o-covar** removes the inter-variable attention module for univariate prediction, causing significant performance decline—the most severe among all variants; **rp-patch** uses standard instead of attention-driven patch partitioning. The experimental results show that the inter-series attention plays an important role in the EPF dataset, and the other components also have a positive impact on the experimental results.

Table 4: Efficiency comparison of different models on EPF-DE dataset.

| Model | Memory (MB) | Training Time / epoch (s) | Inference Time (s) |
|---|---|---|---|
| TiDE | 46.8 | 2.8 | 0.6 |
| Crossformer | 2,315.4 | 10.7 | 1.5 |
| Diffusion-TS | 1,518 | 195 | 62 |
| CSDI | 3,671 | 214 | 65 |
| TMDM | 32,007 | 40.5 | 6 |
| FITS | 199 | 4.9 | 11.6 |

Table 5: Efficiency comparison of different models on Weather dataset.

| Model | Memory (MB) | Training Time / epoch (s) | Inference Time (s) |
|---|---|---|---|
| TiDE | 155.6 | 3.2 | 0.7 |
| Crossformer | 12,798 | 47.2 | 5.2 |
| Diffusion-TS | 1,298 | 197 | 916 |
| CSDI | 21,530 | 270 | 298 |
| TMDM | 15,988 | 22 | 12 |
| FITS | 384 | 7.3 | 45 |

## 5 CONCLUSION

In this work, we propose FITS, an innovative framework that integrates a diffusion generative process with a newly designed transformer-based conditional representation learning framework. In particular, our approach introduces two key innovations: first, we propose an entropy-based adaptive patching method that leverages the sample entropy measure to effectively capture granular local semantics, which avoids information fragmentation caused by discretionary segmentation. Second, we propose a novel cross-variate attention module to effectively capture the evolutionary dynamics of covariates. By using this transformer-based representation module as a conditional guidance for generating future target variables, the diffusion model can be more effectively guided toward the true values. Extensive experiments demonstrate that FITS achieves superior performance in both point forecasting and probabilistic forecasting quality.

Table 6: Ablation experiment results in terms of MSE

| | Wea. | FR | BE |
|---|---|---|---|
| rp-atten | 0.568 | 0.388 | 0.414 |
| w/o-covar | 0.607 | 0.423 | 0.444 |
| rp-patch | 0.569 | 0.391 | 0.401 |
| FITS | **0.559** | **0.384** | **0.398** |

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

## A  TRAINING ALGORITHM

The training procedure is provided in Algorithm 1 below.

---

**Algorithm 1** Training

---

**Require:** Number of diffusion steps $K$.

1: **repeat**
2:     Sample $\mathbf{x}^0_{T:T+H}$ from the training set;
3:     $k \sim \text{Uniform}(\{1, 2, \ldots, K\}), \epsilon \sim \mathcal{N}(0, \mathbf{I})$;
4:     Compute $\mathbf{x}^k_{T:T+H}$ following Eq. (4);
5:     Using the transformer network given in Section 3.2, obtain condition $\mathbf{c}$ based on Eq. (9);
6:     Use the reverse denoising process to generate denoised sample $\mathbf{x}^{k-1}_{T:T+H}$ by Eq. (10);
7:     Calculate the loss $\mathcal{L}_k(\theta)$ in (12);
8:     Take gradient descent step on $\nabla_\theta \mathcal{L}_k(\theta)$;
9: **until** converged

---

## B  DATASETS AND BASELINES

### B.1  DATASETS

We assessed the effectiveness of the proposed FITS model through extensive experiments on 7 time series forecasting datasets. As our focus is on sparse and irregularly sampled time series, we modified the originally regular datasets by applying a subsampling procedure with different filtering rates to induce sparsity.

First, detailed descriptions of the original datasets are provided below:

(1) The **EPF** is an electricity price forecasting dataset, which contains five datasets representing five different day-ahead electricity markets spanning six years each (Lago et al., 2021).

- **NP** represents the Nord Pool electricity market, recording the hourly electricity price, and corresponding grid load and wind power forecast from 2013-01-01 to 2018-12-24.
- **PJM** corresponds to the Pennsylvania - New Jersey - Maryland (PJM) market. It contains the zonal electricity price in the Commonwealth Edison (COMED) area, along with the corresponding system load and COMED load forecast data, spanning from 2013-01-01 to 2018-12-24.
- **BE** stands for Belgium's electricity market. It documents the hourly electricity prices, load forecast in Belgium, and generation forecast in France, covering the period from 2011-01-09 to 2016-12-31.
- **FR** represents the electricity market in France. It records the hourly electricity prices and the corresponding load and generation forecast data, with the time range from 2012-01-09 to 2017-12-31.
- **DE** corresponds to the German electricity market. It keeps track of the hourly electricity prices, the zonal load forecast in the TSO Amprion zone, and the wind and solar generation forecasts, spanning from 2012-01-09 to 2017-12-31.

(2) The **Exchange** (Lai et al., 2018) dataset comprises of daily closing exchange rates of eight currencies against the USD from 1990 to 2016.

(3) The **Weather**(Zhou et al., 2021) dataset contains 21 meteorological variables recorded every 10 minutes at a weather station in Germany during 2020. In this work, we use the Wet Bulb factor as the target variable to be predicted and the other indicators as exogenous variables

Table 7 provides a summary of the data statistics.

Table 7: Full dataset descriptions. Training/Validation/Test dataset is split as 70%/10%/20%.

| Datasets | Look-back period | Forecasting horizon | Target variable | No. of Exogenous Variables | Sampling frequency |
|---|---|---|---|---|---|
| EPF - NP | 192 | 24 | Nord Pool Electricity Price | 2 | 1h |
| EPF - PJM | 192 | 24 | Pennsylvania-New Jersey-Maryland Electricity Price | 2 | 1h |
| EPF - BE | 192 | 24 | Belgium's Electricity Price | 2 | 1h |
| EPF - FR | 192 | 24 | France's Electricity Price | 2 | 1h |
| EPF - DE | 192 | 24 | German's Electricity Price | 2 | 1h |
| Exchange | 96 | 96 | Exchange rates | 7 | 1d |
| Weather | 96 | 96 | $CO_2$-Concentration | 20 | 10m |

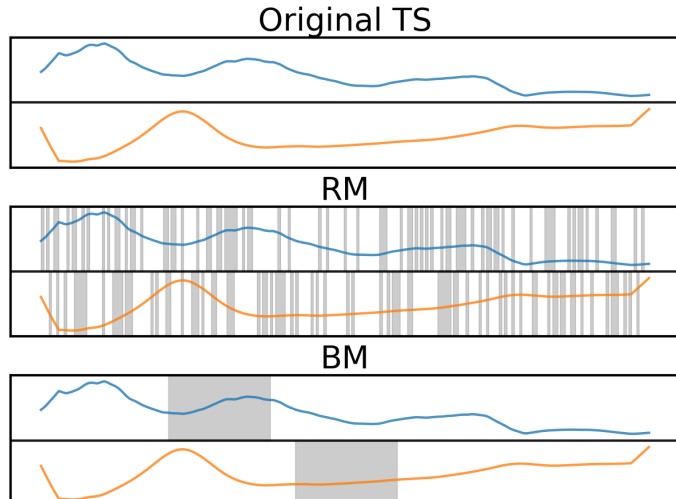

Figure 3: Schematic diagrams of RM and BM, where the gray shaded areas represent the missing regions.

This study employs two distinct downsampling procedures to generate sparse datasets for subsequent model training and inference. The first is a **random missing (RM)** approach, wherein a fraction $\alpha$ of data points is randomly removed from the original target time series, where $\alpha$ is set to 30% and 50%. The forecasting accuracy under different sparsity levels is evaluated in subsequent sections. The second is a **block missing (BM)** approach. For each sliding window, this method removes a continuous segment of length $s$ from a random position within the window. For instance, from a time series segment of length 96, a contiguous segment of 24 points is removed. Figure 3 illustrates examples of the original time series and the sparsified series resulting from these two methods.

## B.2 BASELINES

To comprehensively assess the capabilities of FITS, we benchmark it against state-of-the-art approaches, including time series diffusion models, and other leading methods. This diverse set of baselines ensures a

rigorous and well-rounded comparison, highlighting FITS's performance across different learning paradigms and demonstrating its effectiveness in a wide range of scenarios.

(1) Time series diffusion models:

- CSDI: https://github.com/ermongroup/CSDI. CSDI proposes a novel time series imputation method that leverages score-based diffusion models conditioned on observed data.

- TMDM: https://github.com/LiYuxin321/TMDM. TMDM introduces a Transformer-Modulated Diffusion Model, uniting conditional diffusion generative process with transformers into a unified framework to enable precise distribution forecasting for MTS.

- Diffusion-TS: https://github.com/Y-debug-sys/Diffusion-TS. DiffusionTS is a diffusion model-based framework that decomposes time series into trend, seasonality, and residual components, integrates Transformer architectures to capture temporal dependencies, and aims to produce interpretable and multimodal time series data.

(2) Long time series Forecasting models:

- TiDE:https://github.com/google-research/google-research/blob/master/tide/. TiDE proposes an MLP-based encoder-decoder model for long-term time-series forecasting, which handles covariates and non-linear dependencies.

- DLinear: https://github.com/ioannislivieris/DLinear. DLinear introduces simple one-layer linear models that bypass the temporal information loss inherent in Transformer-based self-attention, achieving superior performance in long-term time series forecasting across diverse datasets.

- Crossformer: https://github.com/Thinklab-SJTU/Crossformer. Crossformer proposes a novel transformer-based model for long-sequence time series forecasting (LSTF), which segments the input into smaller chunks and leveraging cross-attention mechanisms to effectively capture long-range temporal dependencies, thereby enhancing prediction accuracy for extended time horizons.

(3) Irrgular time series Forecasting models:

- mTAN:https://github.com/reml-lab/mTAN. mTAN is a interpolate method which uses learnable time embeddings and an attention mechanism to achieve efficient interpolation and classification.

### B.3 IMPLEMENTATION DETAILS

For all datasets, the pseudo length was fixed at 24. The input sequence length for the EPF dataset was set to 192, yielding a target length of 168, while for the other datasets, the input length was fixed at 96. The forecasting horizon was 24 for EPF and 96 for the remaining datasets. For Quantile Interval Coverage Error(QICE), we divided samples into 10 quantile intervals. For models not inherently designed to handle mismatched lengths between covariates and targets, the target sequences were zero-padded to align dimensions. To enable prediction in models designed for interpolation and imputation, we treat forecasting as a special imputation case where the target value is at the end of the sequence. All implementations were based on PyTorch and executed on an NVIDIA RTX 5090D GPU with 32 GB of memory.

# C    MORE RESULT

## C.1    QUALITATIVE ANYLIST AND VISUALIZATION

We visualized the prediction probabilities of the 0th, 200th, 400th, 600th, and 800th samples in the EPF dataset and compared them with the TMDM model, which performed well in the previous results. It can be clearly seen from the visualization results that our model is more advantageous in terms of the accuracy, concentration of the probability distribution and the fit with the ground truth, which fully demonstrates that our model has a strong ability in predicting probabilities.

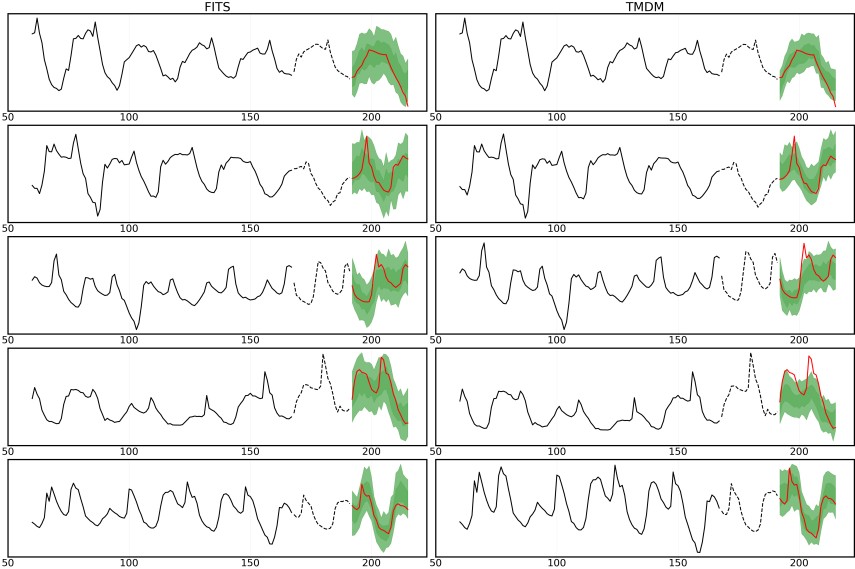

Figure 4: Visualization of PJM Dataset Prediction Results In the visualization, dark green and light green represent the 50% and 90% prediction intervals of the model, respectively, and the red line denotes the ground truth.

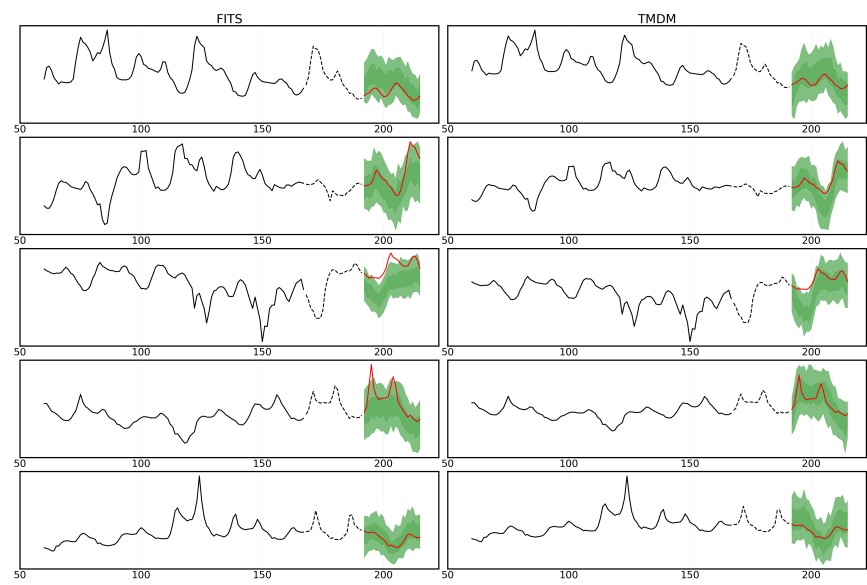

Figure 5: Visualization of NP Dataset Prediction Results In the visualization, dark green and light green represent the 50% and 90% prediction intervals of the model, respectively, and the red line denotes the ground truth.

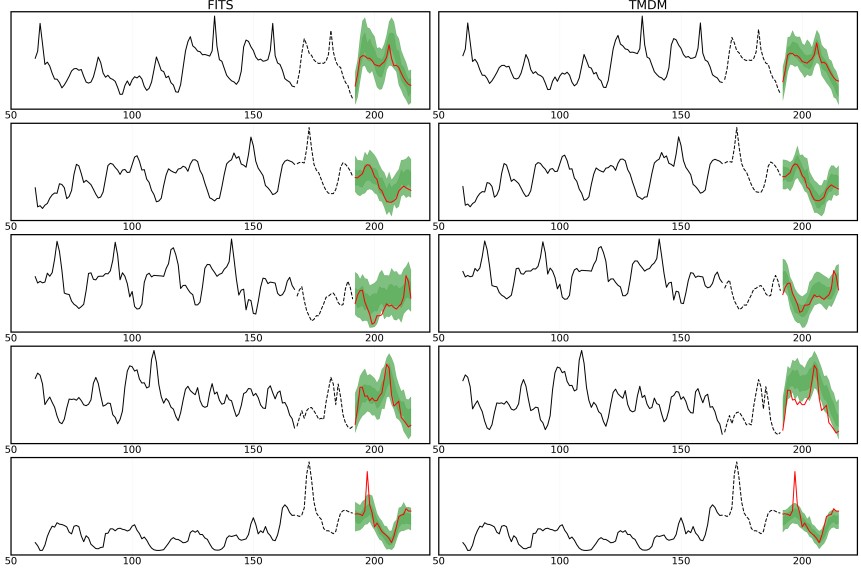

Figure 6: Visualization of FR Dataset Prediction Results In the visualization, dark green and light green represent the 50% and 90% prediction intervals of the model, respectively, and the red line denotes the ground truth.

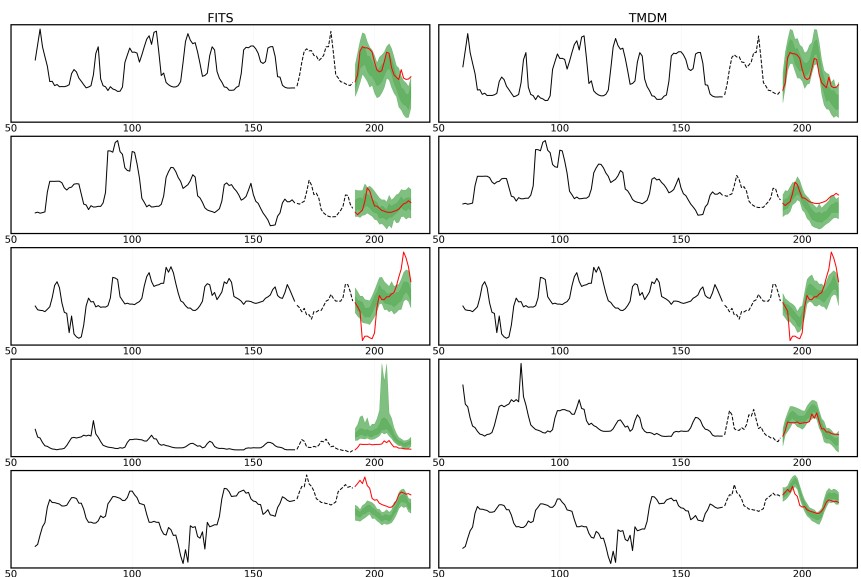

Figure 7: Visualization of DE Dataset Prediction Results In the visualization, dark green and light green represent the 50% and 90% prediction intervals of the model, respectively, and the red line denotes the ground truth.

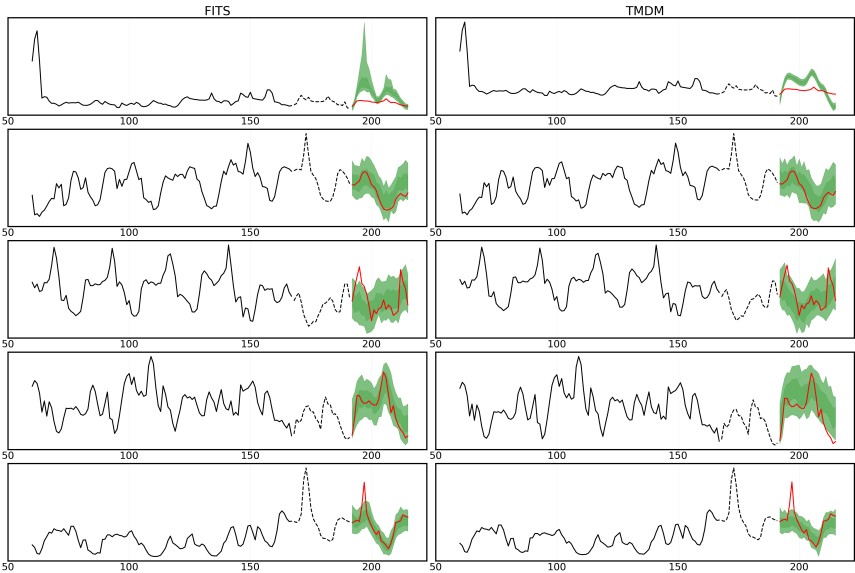

Figure 8: Visualization of BE Dataset Prediction Results In the visualization, dark green and light green represent the 50% and 90% prediction intervals of the model, respectively, and the red line denotes the ground truth.

## C.2 FORECASTING PERFORMANCE WHEN MULTIPLE TARGET VARIABLES ARE PRESENT

To demonstrate our model's ability to simultaneously predict multiple target variables, we selected two best-performing baselines for comparison based on the results presented in Tables 1–3. Specifically, we designated the latter half of the variables in the multivariate dataset as target variables and the first half as covariates. Furthermore, we configured the pseudo-features of all endogenous variables to ensure they have identical lengths.

Table 8: Performance when there are multiple target variables

|  | FR | | BE | | PJM | |
|---|---|---|---|---|---|---|
|  | MSE | MAE | MSE | MAE | MSE | MAE |
| Crossformer | 0.547 | 0.355 | 0.537 | **0.375** | 0.262 | 0.323 |
| DLinear | 0.626 | 0.412 | 0.633 | 0.407 | 0.179 | 0.281 |
| FITS | **0.508** | **0.338** | **0.510** | 0.395 | **0.151** | **0.232** |

## C.3 ENTROPY COMPUTATION PARAMETERS SENSITIVITY ANALYSIS

We conducted experiments on the sensitivity of the parameters in the entropy calculation, and the results are presented in Figure 9. RM is set to 0.3.

In this work, as shown in Section 3.2.1, Sample Entropy is used to compute the information density of each individual patches which is then used to compute the dynamic window boundaries. Specifically, Sample Entropy (SampEn) is a statistical metric used to quantify the complexity and regularity of a time series, the core parameters in its calculation are the similarity tolerance $r$ and the embedding dimension $m$.

Firstly, the Similarity Tolerance $r$ is a threshold for judging whether two m-dimensional reconstructed vectors are "similar", and is usually expressed as a multiple of the standard deviation of the original time series (i.e., $r = \alpha \times$ std, where $\alpha$ is a coefficient). It controls the looseness of similarity judgment and is the most sensitive parameter in SampEn calculation. Empirically, $\alpha$ is set to 0.1 0.25 (i.e., $r = 0.1$std $\sim 0.25$std). In this work, $r$ is set to 0.1, 0.15, 0.2 0.25, 0.3, also keeping other parameters fixed.

Secondly, the embedding dimension $m$ refers to the dimension of reconstructing the original time series into m-dimensional vectors, meaning that $m$ consecutive data points are taken each time to form a reconstructed vector. It determines the granularity of characterizing the local dynamic features of the time series. An excessively small $m$ will lose the multi-dimensional correlation information of the series, while an excessively large $m$ will increase the computational complexity and easily lead to unstable results due to insufficient data volume. According to common practice, $m$ is usually set to 2 or 3. In this work, we demonstrate the performance in terms of MSE for three datasets when $m$ is set to 2, 3, 4, 5, keeping all other parameters fixed.

From Figure 9, we observe that for both parameters $r$ and $m$, the model's MSE performance remains stable across all three datasets. Setting these parameters to appropriate values within the empirical ranges thus has no substantial impact on the overall model performance.

## C.4 MODEL SENSITIVITY TO VARIATIONS IN INITIAL PATCH SIZES $S_{\text{INIT}}$

We assessed the performance across multiple values of initial patch sizes parameter $S_{\text{init}}$ (e.g., 8, 16, 24, 32). The results are shown as Figure 10. Results show that our adaptive scheme maintains stable performance across reasonable $S_{\text{init}}$ ranges, as the BoundaryNet compensates for suboptimal initializations. The final information density quantification and patching effect are dominated by adaptive adjustment, rather than the initial $S_{\text{init}}$.

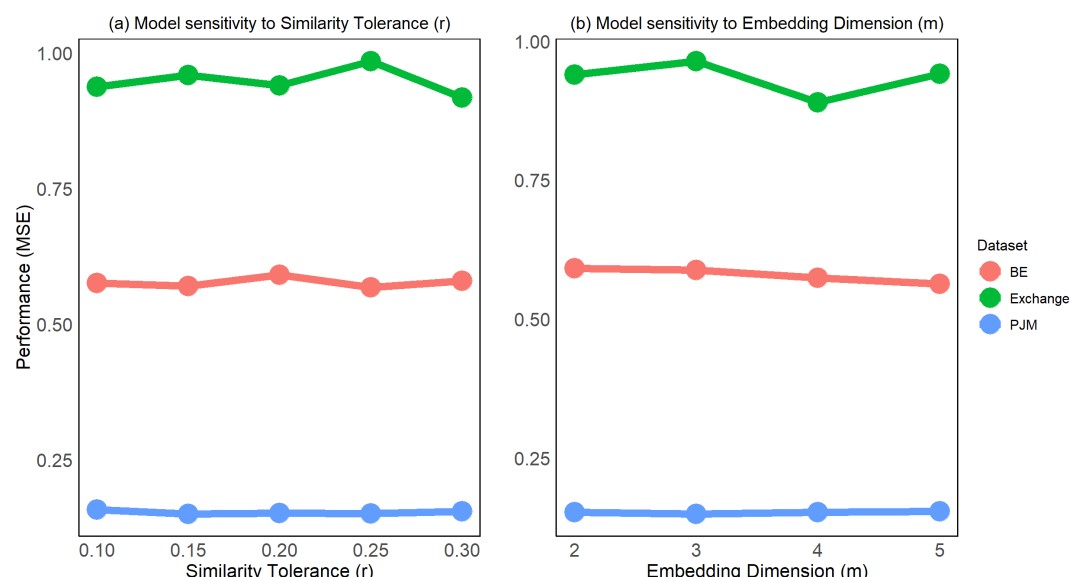

Figure 9: Model performance across (a) similarity tolerance $r = \{0.1, 0.15, 0.2, 0.25, 0.3\}$, and (b) embedding dimension $m = \{2, 3, 4, 5\}$. The relatively flat curves in the graph indicate that our model is not sensitive to variations in $m$ and $r$.

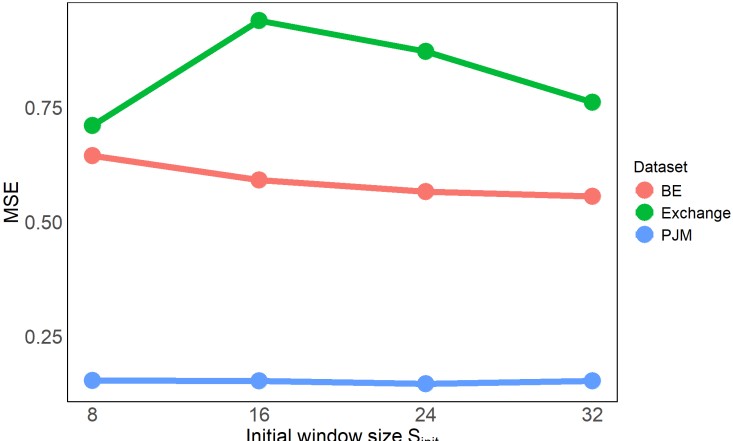

Figure 10: Model sensitivity to variations in initial patch size parameter $S_{\text{init}}$

## D  THE USE OF LARGE LANGUAGE MODELS (LLM)

This work used LLMs to fix grammar mistakes and spelling errors.

