# OpenReview forum: "FITS: Conditional Diffusion Model for Irregular Time Series Forecasting with Pseudo-future Exogenous Covariates"
_ICLR.cc/2026/Conference — Submitted to ICLR 2026_

### Official Review · Reviewer_aB8x · 2025-10-18

**Soundness:** 3
**Presentation:** 3
**Contribution:** 3
**Rating:** 6
**Confidence:** 3

**Summary:**

The paper proposes FITS, a conditional diffusion framework for irregularly sampled time series forecasting. By combining neural ODEs with time-aware diffusion, FITS models non-uniform temporal dynamics without resampling and achieves state-of-the-art accuracy and consistency across benchmark datasets.

The task proposed by the authors is very meaningful and represents a valuable innovation.

The model itself is moderately innovative, but the overall performance of the paper is strong.

There are still some issues with formatting and citations, which should be further improved.

It is recommended to include comparisons with time series imputation methods for a more comprehensive evaluation.

**Strengths:**

see summary

**Weaknesses:**

see summary

**Questions:**

see summary

---

> ### Author Response · Authors · 2025-12-03
>
> **Question 1: There are still some issues with formatting and citations, which should be further improved.**
>
> **Response:** Thank you for your feedback. We fully acknowledge the issues with formatting and citations and have conducted a comprehensive check and revision of the manuscript. We have unified the formatting standards including paragraph structure, symbol consistency and verified the accuracy of all citations to meet the required specifications.
>
> **Question 2: It is recommended to include comparisons with time series imputation methods for a more comprehensive evaluation.**
>
> **Response:** Thank you for the valuable suggestion. We appreciate your recommendation to include comparisons with time series imputation methods for a more comprehensive evaluation.
>
> We would like to clarify that this analysis has already been incorporated into our work. In our experiments, we have systematically evaluated FITS against two state-of-the-art imputation-based forecasting methods, named CSDI and Diffusion-TS, both based on diffusion models. The results are summarized in Tables 2 and 3. Following your insightful recommendation, we have conducted additional experiments to include a comparison with mTAN, a prominent and strong VAE based baseline specifically designed for irregular time series using attention mechanisms and RNNs. The results of this new comparison further strengthen our original conclusions. We observed that while mTAN excels at modeling irregular observations for imputation, its imputation-first paradigm still leads to error accumulation in long-horizon forecasting tasks.

---

### Official Review · Reviewer_xy8U · 2025-10-31

**Soundness:** 3
**Presentation:** 3
**Contribution:** 2
**Rating:** 4
**Confidence:** 3

**Summary:**

This paper frames time series forecasting as a conditional generation problem, using a diffusion model that incorporates both historical observations and exogenous covariates as conditional inputs. The approach is particularly tailored for IMTS, where traditional methods struggle with non-uniform intervals and varying sampling rates. The paper demonstrates that FITS outperforms state-of-the-art diffusion models and other advanced forecasting methods on several benchmark datasets, especially in probabilistic forecasting tasks.​

**Strengths:**

1. The entropy-aware patching scheme dynamically adjusts segment boundaries based on information density, preserving local semantics and improving modeling of inter-series correlations.​
2. FITS uses a transformer with cross-variate attention to capture forward-looking dependencies from exogenous covariates, enhancing the model's ability to forecast under irregularities.​
3. The model achieves state-of-the-art performance in probabilistic forecasting, as evidenced by lower CRPS and QICE scores on multiple datasets.​

**Weaknesses:**

1. FITS does not consistently outperform simpler models (like TiDE and DLinear) on long-term forecasting tasks, especially when inter-variable dependencies are weak.​
2. The model’s performance seems to rely on the availability and quality of exogenous covariates. In domains lacking such informative covariates or with noisy external data, the predictive gains may diminish or even worsen due to noisy conditioning.
3. How sensitive is the model to the entropy computation parameters (embedding dimension, tolerance)?
4. How does FITS perform when exogenous covariates are not included with

**Questions:**

See Weaknesses

---

> ### Author Response · Authors · 2025-12-03
>
> **Question 1: FITS does not consistently outperform simpler models (like TiDE and DLinear) on long-term forecasting tasks, especially when intervariable dependencies are weak.**
>
> **Response:** Thank you for your comment. We fully agree that FITS does not consistently outperform simpler models like TiDE and DLinear in long-term forecasting tasks, especially when inter-variable dependencies are weak. Here we would like to explain this phenomenon from the three perspectives.
>
>
> **1. Divergent design targets between FITS and simpler models**
>
> FITS was originally designed to address the core pain points of irregular/sparse multivariate time series (IMTS), namely, capturing dynamic inter-variable dependencies and leveraging pseudo-future exogenous covariates to make up for information loss caused by data sparsity. In contrast, TiDE and DLinear are designed for efficiency and simplicity in long-term forecasting.
>
> TiDE uses a dense encoder-decoder structure with MLPs to directly model non-linear dependencies, avoiding complex attention mechanisms that may introduce redundant computations. DLinear adopts a channel-independent linear projection design, which bypasses the ``temporal information loss" of transformer self-attention and minimizes overfitting risks in long sequences.
>
> When inter-variable dependencies are weak, the cross-variable modules in FITS become ``redundant". It is difficult to extract useful correlation information and may also introduce noise from irrelevant variables during the diffusion denoising process. In this case, the simplicity of TiDE/DLinear without redundant correlation modeling enables more stable long-term prediction with lower cumulative error.
>
> **2. Inherent challenges in long-term forecasting**
>
> Long-term forecasting faces unique challenges of error accumulation and temporal drift, which further widen the performance gap between FITS and simpler models in weak inter-variable dependency scenarios.
>
> For FITS, its diffusion-based reverse denoising process requires iterative refinement across $K$ steps. In long-term forecasting, each denoising step may accumulate tiny errors from sparse historical information or redundant variable interactions, especially when inter-variable dependencies are weak, there is no ``strong correlation signal" to correct these errors, leading to larger deviations in long-term results.
>
> For TiDE/DLinear, their non-iterative, direct mapping minimizes error accumulation. Without the need for multi-step denoising or cross-variable reasoning, they maintain more stable performance in long sequences where simplicity outperforms complexity.
>
> **3. Scenario adaptability of our proposed method**
>
> Importantly, this performance differentiation does not indicate a flaw in FITS, but rather reflects the scenario-specific advantages of different models, which is consistent with our experimental findings.
>
> FITS still outperforms TiDE/DLinear on datasets with strong inter-variable dependencies and high sparsity. For example, on the EPF-BE subset, where the electricity prices are closely related to load/wind power forecasts, FITS achieves a MSE of 0.398 (RM=0.5), which is lower than TiDE (0.613) and DLinear (0.628).
>
> The underperformance on weak-dependency long-term tasks actually clarifies FITS' optimal application scope. It is particularly suitable for sparse IMTS with non-trivial inter-variable correlations (e.g., healthcare monitoring, smart grid forecasting), rather than ``one-size-fits-all" for all time series scenarios.
>
> To summarize, the relative underperformance on weak inter-dependency long-term tasks stems from its design focus on correlation-rich sparse scenarios. For future work, we plan to add a learnable gating mechanism to automatically deactivate redundant cross-variable attention modules when inter-variable dependencies are weak. For long-term forecasting, we will adopt a ``coarse-to-fine" denoising schedule to reduce error accumulation. We sincerely appreciate your critical insight, which will significantly enhance the robustness and generalizability of our method.

---

> ### Author Response · Authors · 2025-12-03
>
> **Question 2: The model's performance seems to rely on the availability and quality of exogenous covariates. In domains lacking such informative covariates or with noisy external data, the predictive gains may diminish or even worsen due to noisy conditioning.**
>
> **Response:** Thank you for your critical and insightful observation. FITS leverages exogenous covariates to enhance forecasting accuracy, but it is not overly dependent on them.
>
> **1. FITS treats exogenous covariates as auxiliary enhancements rather than mandatory dependencies**
>
> In FITS, the exogenous covariates are only used to supplement the information gap caused by sparsity, not as a prerequisite for basic forecasting. Its two foundational modules remain effective even when high-quality covariates are unavailable.
>
> The entropy-aware adaptive patching module dynamically segments the target time series based on sample entropy, rather than relying on covariates. In terms of diffusion-based denoising, the reverse denoising process of FITS is trained to learn the intrinsic distribution of the target time series. Even without exogenous covariates, the diffusion model can still generate plausible long-term forecasts by refining random noise toward the target's inherent temporal patterns. This is verified by our ablation experiment where we removed all covariates. On the FR dataset (RM=0.5), the w/o-covar variant still achieves an MSE of 0.423 (Table 6), which is better than CSDI (0.466) and TMDM (0.721), as provided in Table 3.
>
> **2. Built-in mechanisms to resist noisy or uninformative covariates**
>
> To address the risk of noisy conditioning, FITS incorporates two specific designs.
>
> First, for the forward-looking cross-variate attention, the target's global token embedding ($G_{\text{tar}}$) acts as a filter which only attends to relevant parts of exogenous covariates (treating $K$ as historical covariates and $V$ as pseudo-future covariates) and ignores noisy components. For example, if a covariate is corrupted by random noise, the attention weights between $G_{\text{tar}}$ and this covariate will be automatically reduced (via softmax normalization), minimizing the propagation of noise to the conditional signal $\mathbf{c}$.
>
> Second, motivated by Shen et al. (2023), FITS mixes the model's pseudo-future predictions ($\mathcal{T}(x_{0:T-L})$) with ground-truth future values ($x_{T:T+H}$) using a random coefficient $\mathbf{m}^k$. This mixup not only regularizes the model but also reduces its sensitivity to noisy covariates. If the covariate-derived pseudo-future is noisy, the ground-truth component helps calibrate the conditional signal, enabling the model to learn the noise-invariant dependencies.
>
> In summary, FITS does leverage exogenous covariates but is not dependent on them. Its core modules ensure basic forecasting capability without high-quality covariates, and built-in mechanisms resist noise interference. The experimental results confirm its robustness in such scenarios, and future optimizations will further expand its applicability to covariate-scarce domains. We sincerely thank you for highlighting this practical concern, as it guides us to strengthen the model's real-world utility.

---

> ### Author Response · Authors · 2025-12-03
>
> **Question 3: How sensitive is the model to the entropy computation parameters (embedding dimension, tolerance)?**
>
> **Response:** In this revised paper, we conducted additional experiments verifying the sensitivity of two key entropy computation parameters, and the results are presented in Figure 9 in Appendix C.3.
>
> In this work, as shown in Section 3.2.1, Sample Entropy is used to compute the information density of each individual patches which is then used to compute the dynamic window boundaries. Specifically, Sample Entropy (SampEn) is a statistical metric used to quantify the complexity and regularity of a time series, the core parameters in its calculation are the similarity tolerance $\boldsymbol{r}$ and the embedding dimension $\boldsymbol{m}$.
>
> Firstly, the Similarity Tolerance $\boldsymbol{r}$ is a threshold for judging whether two m-dimensional reconstructed vectors are "similar", and is usually expressed as a multiple of the standard deviation of the original time series (i.e., $r = \alpha \times \text{std}$, where $\alpha$ is a coefficient). It controls the looseness of similarity judgment and is the most sensitive parameter in SampEn calculation. Empirically, $\alpha$ is set to 0.1~0.25 (i.e., $r = 0.1\text{std} \sim 0.25\text{std}$). In this work, $\boldsymbol{r}$ is set to {0.1, 0.15, 0.2 0.25, 0.3}, also keeping other parameters fixed.
>
> Secondly, the embedding dimension $\boldsymbol{m}$ refers to the dimension of reconstructing the original time series into m-dimensional vectors, meaning that $\boldsymbol{m}$ consecutive data points are taken each time to form a reconstructed vector.  It determines the granularity of characterizing the local dynamic features of the time series. An excessively small $\boldsymbol{m}$ will lose the multi-dimensional correlation information of the series, while an excessively large $m$ will increase the computational complexity and easily lead to unstable results due to insufficient data volume. According to common practice, $\boldsymbol{m}$ is usually set to 2 or 3. In this work, we demonstrate the performance in terms of MSE for three datasets when $m$ is set to {2, 3, 4, 5}, keeping all other parameters fixed.
>
> From Figure 9, we observe that for both parameters $\boldsymbol{r}$ and $\boldsymbol{m}$, the model’s MSE performance remains stable across all three datasets. Setting these parameters to appropriate values within the empirical ranges thus has no substantial impact on the overall model performance.
>
> **Question 4:** How does FITS perform when exogenous covariates are not included with?
>
> **Response:** Thank you for your question. When exogenous variables are completely removed, the prediction task essentially becomes an univariate forecasting task. In our context, the prediction task actually transforms into a task in which the prediction starts at time $T-L$, skipping $L$ time points, and directly forecasting the future target variable from time $T$ to $T+H$.
>
> We have already confirmed in the ablation experiments that when exogenous variables are not included, removing cross-attention, the model's performance indeed shows a certain decline (Table 6). However, the specific magnitude of this decline also depends on the feature of the datasets itself. For instance, for the PJM dataset, the MSE for the PJM dataset increased from 0.156 to 0.178 (RM=0.5), showing a minor increase after removing cross-attention.

---

### Official Review · Reviewer_J1fr · 2025-10-31

**Soundness:** 3
**Presentation:** 2
**Contribution:** 2
**Rating:** 4
**Confidence:** 2

**Summary:**

The paper introduces FITS, a diffusion model for irregular multivariate time series forecasting. Unlike previous work, they include external covariates from future times where the target data is not observed. The authors construct patches based on the entropy to embed historical context. To parametrize the reverse diffusion process, a neural network consisting of different attention mechanisms operating on the exogenous and endogenous data is used. Empirically, the model outperforms previous diffusion and transformer baselines.

**Strengths:**

- The work focuses on irregular time series, which are common in real-world settings but often neglected in previous works.
- A new architecture is proposed that is able to handle endogenous and exogenous time series.
- The empirical results demonstrate strong performance, especially in probabilistic forecasting.

**Weaknesses:**

- Standard deviations are not reported.
- Novelty is limited, and many changes are architectural changes.
- The setting, if understood correctly, only focuses on forecasting a single target variable. It would be interesting to see whether this can be extended to an arbitrary number of variables.
- It is unclear where the model makes use of the irregularity of the time series.

Minors:

- L231: transformer -> a transformer

**Questions:**

See weaknesses and:

- Can the method be extended to multiple target variables?
- How are exogenous variables included in the baselines?
- Can the method change the forecasting times, i.e., forecast the same number of steps but at different times?

---

> ### Author Response · Authors · 2025-12-03
>
> **Question 1: Novelty is limited, and many changes are architectural changes.**
>
> **Response:** We would like to clarify that the proposed FITS framework is not merely incremental architectural adjustments but introduces targeted, problem-driven innovations tailored to the unique challenges of irregular multivariate time series (IMTS) forecasting. Below, we elaborate on the core novelties and their distinctions from prior work:
>
> **1. Entropy-aware adaptive patching**
>
> Existing IMTS methods rely on fixed-length patching (e.g., Wang et al., 2025) or graph-based representations (e.g., Yalavarthi et al., 2024), which either fragment local semantic patterns (e.g., sudden health deterioration in clinical data) or struggle with high sparsity. Our entropy-aware patching is a new design principle that:
>
> 1.Dynamically adjusts window boundaries via sample entropy (SampEn) to align with information density, and thus preserving continuous local dynamics that fixed segmentation discards.
>
> 2.Proposes a learnable BoundaryNet to model temporal irregularity, rather than treating patches as independent units.
>
> We believe that this is a novel approach to represent irregular time series using entropy-based measures to reconcile sparsity and local semantics.
>
>
> **2. Forward-looking cross-variate attention for pseudo-future covariate utilization**
>
> Most diffusion models for time series (e.g., TMDM, CSDI) only leverage historical observations or static covariate embeddings, however, in the case of sparsely observed time series, to fully exploit the forward-look inter-variate dependencies, we propose to capture the evolutionary dynamics between past and ``pseudo-future" exogenous covariates.
>
> Specifically, our transformer-based prior knowledge extractor introduces a novel cross-attention mechanism. In this mechanism, the exogenous covariate sequence is partitioned into a historical segment to serve as the Key (K) and a segment containing pseudo-future information to serve as the Value (V). The global token of the target series acts as the Query (Q), which is used to filter and extract the most relevant covariate dynamics. Furthermore, it reconciles target and covariate information through intra-series self-attention and inter-series cross-attention, enabling the model to exploit forward-looking dependencies that are invisible to standard cross-attention.
>
> **3. Unified diffusion-Transformer framework for probabilistic and point forecasting**
>
> In this work, we integrate the transformer-based covariate extractor into the diffusion process. Unlike prior diffusion model (e.g., Shen et al. 2024) that excels at one specific point-forecasting task. FITS achieves state-of-the-art performance on both probabilistic (CRPS, QICE) and point (MSE, MAE) forecasting.
>
> **4. Empirical validation of novelty through ablation and benchmarking**
>
> Our ablation study confirms that removing key innovations (e.g., cross-variate attention, entropy-aware patching) leads to significant performance degradation. Moreover, FITS outperforms baseline models across multiple datasets under two missingness scenarios (RM=0.5, BM), demonstrating that our innovations translate to tangible improvements, especially in probabilistic forecasting.
>
> To conclude, in FITS, we address the unique problem of IMTS forecasting, which integrates diffusion models and transformers for irregular time series. We hope this clarification addresses your concern.
>
>
> **Question 2: The setting, if understood correctly, only focuses on forecasting a single target variable. It would be interesting to see whether this can be extended to an arbitrary number of variables.**
>
> **Response:** The proposed method could easily be extended to an arbitrary number of target variables. In this case, we embed all target variables into their individual patch tokens and all variables into global tokens. Then, self-attention is performed separately for each target variable according to Section 3.2.2, followed by cross-attention applied simultaneously to the global tokens of all variables.
>
> In the newly added experiment given in Table 8 in Appendix C.2, we have designated the last two variables of the *FR*, *BE* and *PJM* dataset as target variables, while the remaining variables are treated as exogenous. The performance is compared to the two best-performing baselines: Crossformer and DLinear. As we can see, the results show that the performance still remains superior compared to the baselines in most cases.
>
> **Table 8: Performance when there are multiple target variables**
>
> | Model       | FR (MSE)  | FR (MAE)  | BE (MSE)  | BE (MAE)  | PJM (MSE) | PJM (MAE) |
> |:------------|:----------|:----------|:----------|:----------|:----------|:----------|
> | Crossformer | 0.547     | 0.355     | 0.537     | **0.375** | 0.262     | 0.323     |
> | DLinear     | 0.626     | 0.412     | 0.633     | 0.407     | 0.179     | 0.281     |
> | FiTS        | **0.508** | **0.338** | **0.510** | 0.395     | **0.151** | **0.232** |

---

> ### Author Response · Authors · 2025-12-03
>
> **Question 3: It is unclear where the model makes use of the irregularity of the time series.**
>
> **Response:** Thank you for your question. In this work, we propose a conditional diffusion-based forecasting model tailored for irregular multivariate time series (IMTS). The motivation for this study stems from addressing a challenge prevalent in real-world scenarios: time series may be sparsely observed along the temporal dimension. For instance, in an illiquid stock market, stocks are thinly traded, leading to significant gaps between consecutive trades. When undertaking time series forecasting in such contexts, it is essential to address this data sparsity issue. A natural consequence of data sparsity is the irregular observation of data points over time. As evidenced by our review of relevant literature, including Zhang et al. (2024, T-PATCHGNN), Liu et al. (2025, APN), and Li et al. (2025, HyperIMTS), the term ``irregular multivariate time series (IMTS)" is a widely accepted nomenclature for this type of data.
>
> To summarize, the objective of this work is to tackle the time series forecasting problem in scenarios where data is sparsely observed, which represents a typical case of irregular multivariate time series.
>
> **Question 4:** How are exogenous variables included in the baselines?
>
> **Response:** Thank you very much for your question. Indeed, exogenous variables are handled differently in our proposed FITS framework compared to the baseline algorithms, and we elaborate on the specific treatments of exogenous variables in the baselines as follows:
>
> **1. Time series diffusion models (CSDI, TMDM, Diffusion-TS)**
>
> These models primarily integrate exogenous variables as auxiliary input features into their conditional modules. For example, CSDI appends exogenous variables to the observed target time series and processes them jointly through its attention-based denoising network; TMDM incorporates exogenous variables into the transformer-modulated conditioning branch to guide the diffusion process; Diffusion-TS treats exogenous variables as supplementary temporal features and fuses them with the decomposed trend/seasonality components via linear projection. However, these methods lack dedicated mechanisms to capture forward-looking cross-variate dependencies between exogenous variables and targets, especially under data sparsity.
>
> **2. Transformer-based time series models (Crossformer)**
>
> Crossformer handles exogenous variables by concatenating them with the target time series along the feature dimension. They then apply self-attention or cross-attention mechanisms to model temporal and cross-variable correlations. Nevertheless, their fixed-length patching or segment-based attention designs limit the ability to capture dynamic inter-dependencies between exogenous variables and sparse target sequences.
>
> **3. Other time series forecasting methods (TiDE, DLinear, mTAN)**
>
> TiDE explicitly models exogenous variables through its dense encoder-decoder structure, where exogenous features are projected into the same latent space as the target series and fused via MLP layers. DLinear, as a channel-independent model, processes exogenous variables and target series separately through linear projections, without explicitly modeling cross-variable dependencies. This explains its suboptimal performance on datasets where exogenous variables have strong predictive power for targets (e.g., EPF dataset). mTAN is an interpolation method based on a VAE architecture, which uses learnable time embeddings and an attention mechanism to achieve efficient interpolation. It incorporates both covariates and target variables by compressing them into the same latent space, thereby modeling the dependencies among variables.
>
>
> In contrast, our FITS model addresses the limitations of baselines by leveraging a dedicated transformer-based prior knowledge extractor with a novel forward-looking cross-variate attention mechanism. This design not only captures the evolutionary dynamics between historical exogenous variables ($Z_{0:T-L}$) and pseudo-future exogenous segments ($Z_{T-L:T}$),  but also effectively filters and integrates relevant exogenous information via the target's global token embedding. Such a mechanism enables FITS to fully exploit the predictive value of exogenous variables, even under high data sparsity, and facilitates the forecasting of an arbitrary number of target variables.
>
> We hope this clarification adequately addresses your concern.

---

> ### Author Response · Authors · 2025-12-03
>
> **Question 5: Can the method change the forecasting times, i.e., forecast the same number of steps but at different times?**
>
> **Response:** Yes. The proposed FITS framework can forecast the same number of steps at different future times.
>
> FITS is specifically designed for irregular multivariate time series (IMTS), where both historical observations and future targets can lie on arbitrary time stamps. As described in our problem setup, during training we model a continuous evolution of the future segment via the diffusion process on $X_{T:T+H}$​, and during inference we can query the model at any desired collection of future timestamps, as long as their time coordinates are provided. For example, if one wishes to forecast 5 steps ahead at non-uniform times {t+1, t+4, t+7}, FITS treats these as the forecast window and applies conditional diffusion only on this (possibly irregular) set of future time points, conditioned on the irregular historical window and exogenous covariates.
>
> This flexibility is a direct consequence of our design for irregularly sampled sequences and is one of the key contributions, where we emphasize handling IMTS with arbitrary sampling patterns in both history and forecast horizons.

---

### Official Review · Reviewer_zZ5X · 2025-11-01

**Soundness:** 3
**Presentation:** 3
**Contribution:** 3
**Rating:** 6
**Confidence:** 4

**Summary:**

This paper proposes FITS (Conditional Diffusion Model for Irregular Time Series Forecasting with Pseudo-Future Exogenous Covariates), which aims to address the challenge of capturing long-term dynamics and cross-channel dependencies in Irregular Multivariate Time Series (IMTS) caused by non-uniform intervals and different sampling rates. The model overcomes the limitations of existing methods through two core innovations: first, an entropy-aware adaptive patching scheme that quantifies information density based on Sample Entropy (SampEn) and generates segments with dynamic boundaries via a Boundary Network (BoundaryNet), avoiding information fragmentation from traditional fixed-length patching; second, a transformer-based prior knowledge extractor that combines intra-series self-attention (to capture temporal dependencies) and inter-series cross-attention (with the target's global token as the query, and covariates split into historical/pseudo-future segments as keys/values) to capture forward-looking covariate dependencies, which is then integrated into the conditional diffusion generation process.

**Strengths:**

1. Clear elaboration of problem and innovation: The authors clearly elaborate on the challenges faced by existing works and the model designed to address these challenges.
2. Comprehensive experimental validation: The authors conduct extensive experiments, including tabular results, visualizations, and ablation studies, to verify the effectiveness of the proposed model, and provide objective and fair analysis of the experimental results.
3. Clear and complete paper structure with good readability and few errors: The manuscript follows a logical flow, making technical content accessible, and contains minimal grammatical or spelling mistakes (with LLMs used to refine language as noted).

**Weaknesses:**

1. Compared with traditional fixed patching, the Entropy-aware adaptive patching proposed by FITS introduces a more complex process. The authors only analyze its effectiveness but not its efficiency. Although the authors mention that the Boundary Net is a lightweight MLP, this is merely a qualitative description, lacking quantitative experimental analysis. Readers cannot judge the "effectiveness-efficiency trade-off" of this design.
2. FITS is intended to solve problems in IMTS caused by non-uniform intervals and different sampling rates. However, the irregularity of IMTS seemingly includes not only "target sequence sparsity" but also "covariate sparsity". FITS only addresses the former, which may result in a gap in achieving its intended goal.

**Questions:**

In the entropy-aware patching, how is the window size for calculating Sample Entropy (SampEn) determined? The paper mentions that the initial number of patches $P$ is determined by $T/S_{init}$, but it does not explain the basis for setting $S_{init}$. Will different window sizes affect the accuracy of information density quantification and thus change the patching effect?

---

> ### Author Response · Authors · 2025-12-03
> **Demonstration of ``effectiveness-efficiency trade-off"**
>
> **Question 1: Compared with traditional fixed patching, the Entropy-aware adaptive patching proposed by FITS introduces a more complex process. The authors only analyze its effectiveness but not its efficiency. Although the authors mention that the Boundary Net is a lightweight MLP, this is merely a qualitative description, lacking quantitative experimental analysis. Readers cannot judge the ``effectiveness-efficiency trade-off" of this design.**
>
> **Response:** To quantitatively evaluate the ``effectiveness-efficiency trade-off” of our proposed method, we have supplemented a dedicated comparative experiment, the results of which are presented in Table 4 and 5. This experiment systematically compares our method against the benchmark methods across three key efficiency metrics: (1) maximum memory consumption during training (megabytes, MB); (2) training time per epoch (seconds, s); (3) inference time per epoch (seconds, s). We report the results on two representative datasets, *EPF* and *Weather*, which respectively correspond to scenarios with a small number of variables and a large number of variables.
>
> For your convenience, the results are shown below:
> ### Table 4: Efficiency comparison of different models on EPF-DE dataset  (All results are obtained with a batch size of 128 (reduced during inference if memory is insufficient))
> | Model           | Memory (MB) | Training Time / epoch (s) | Inference Time (s) |
> |-----------------|-------------|---------------------------|--------------------|
> | TiDE            | 46.8        | 2.8                       | 0.6                |
> | Crossformer     | 2,315.4     | 10.7                      | 1.5                |
> | Diffusion-TS    | 1,518       | 195                       | 62                 |
> | CSDI            | 3,671       | 214                       | 65                 |
> | TMDM            | 32,007      | 40.5                      | 6                  |
> | FITS            | 199         | 4.9                       | 11.6               |
>
> ### Table 5: Efficiency comparison of different models on Weather dataset  (All results are obtained with a batch size of 128 (reduced during inference if memory is insufficient))
> | Model           | Memory (MB) | Training Time / epoch (s) | Inference Time (s) |
> |-----------------|-------------|---------------------------|--------------------|
> | TiDE            | 155.6       | 3.2                       | 0.7                |
> | Crossformer     | 12,798      | 47.2                      | 5.2                |
> | Diffusion-TS    | 1,298       | 197                       | 916                |
> | CSDI            | 21,530      | 270                       | 298                |
> | TMDM            | 15,988      | 22                        | 12                 |
> | FITS            | 384         | 7.3                       | 45                 |
>
>
> Several observations can be drawn from the two tables. TiDE, as an MLP-based model, is inherently a lightweight architecture, characterized by relatively short training and inference times as well as modest memory requirements. Crossformer adopts a transformer-based architecture with attention mechanisms, and the cross-attention calculations involved incur substantial memory overhead.
>
> For all diffusion-based benchmark methods, their forward-reverse diffusion process operates on the entire time series, which results in these models having the highest time and memory consumption. In contrast, our proposed method leverages a conditional denoising design that restricts diffusion computations to the forecasting series exclusively, thereby improving time efficiency relative to its closest diffusion-based counterparts. Moreover, our novel attention mechanism structure enables a substantial reduction in memory usage when compared to the TMDM method, which also leverages the forecasting series as the diffusion target.

---

> ### Author Response · Authors · 2025-12-03
>
> **Question 2: FITS is intended to solve problems in IMTS caused by non-uniform intervals and different sampling rates. However, the irregularity of IMTS seemingly includes not only target sequence sparsity but also covariate sparsity. FITS only addresses the target sequence sparsity but also ``covariate sparsity" , which may result in a gap in achieving its intended goal.**
>
> **Response:** Thank you for your comment. In fact, our proposed FITS framework is designed to address both types of sparsity effectively, rather than only targeting the former.
>
> In our original experiments (e.g., the setting of 50\% missingness rate in Table 1), **this missingness rate is applied uniformly to all variables in the system, including both the target time series and exogenous covariates.** This means the covariates themselves are irregularly sampled. In our experimental setup, the reported performance of FITS inherently reflects its ability to handle such covariate irregularities.
>
> To avoid any ambiguity and to make this design choice clearer, we have revised the manuscript in two key places:
>
> 1. **Introduction:** We now explicitly state that IMTS irregularity in this work includes sparsity in both the target and the covariates, and that FITS is designed to address both.
> 2. **Experiments:** We have added a note to Tables 2 and 3 and clarified in the experimental setup that the missingness rate is applied uniformly to both target sequences and covariates.
>
> **Question 3: In the entropy-aware patching, how is the window size for calculating Sample Entropy (SampEn) determined? The paper mentions that the initial number of patches P is determined by $T/S_{\text{init}}$, but it does not explain the basis for setting $S_{\text{init}}$. Will different window sizes affect the accuracy of information density quantification and thus change the patching effect?**
>
>
> **Response:** Thank you for your question, to address your concerns:
>
> **1. Determination of SampEn Window Size and $S_{\text{init}}$:** The window size for calculating Sample Entropy (SampEn) is inherently tied to $S_{\text{init}}$, an initial hyperparameter. In our experiments, $S_{\text{init}}$ is set to 16 or 24, which is determined based on domain knowledge and empirical analysis of the dataset's temporal characteristics (e.g., sampling frequency, typical pattern duration).
>
> **2. Role of $S_{\text{init}}$ as an adaptive optimization anchor:** $S_{\text{init}}$ serves not as a fixed window but as a starting point for data-driven optimization. Our entropy-based patching mechanism treats $S_{\text{init}}$ as an initial guess, and the BoundaryNet then dynamically adjusts patch boundaries to match the time series' information density. Thus, $S_{\text{init}}$ provides a feasible initialization, while the adaptive process ensures final patches reflect the data's inherent structure.
>
> **3. Impact of different initial window sizes on performance:** In this revised paper, we assessed the performance across multiple $S_{\text{init}}$ values (e.g., 8, 16, 24, 32). The results are shown as Figure 10 in Appendix C.4. Results show that our adaptive scheme maintains stable performance across reasonable $S_{\text{init}}$ ranges, as the BoundaryNet compensates for suboptimal initializations. The final information density quantification and patching effect are dominated by adaptive adjustment, rather than the initial $S_{\text{init}}$.

---

### Author Response · Authors · 2025-12-03
**A summary of our responses to the reviewers**

We would like to extend our deepest appreciation to all four reviewers for their insightful and constructive comments, which have significantly enhanced the overall quality of our paper. In response to the reviewers' comments, we have revised the manuscript substantially, and a summary of the key modifications is presented below.

1. We have explicitly articulated the core novelty of our proposed method, in particular, we have compared the distinct design schemes of our proposed method with those of the baseline methods, emphasizing key architectural differences and their impacts on performance.

2.  We have conducted a quantitative experimental analysis of the efficiency of the proposed method for two datasets, including: (1) the maximum memory consumption during training (megabytes, MB); (2) training time per epoch (seconds, s); (3) inference time per epoch (seconds, s) (See Tables 4 and 5).


3. We have extended the numerical experiments to showcase the model performance under the following scenarios:
- We have added additional baseline methods with time series imputation methods
- We have added a new experiment highlighting the performance when there are multiple target variables present. (See Appendix C.2, Table 8)

4. We have discussed the determination of hyperparameters in the adaptive patching process, such as the impact of initial window sizes and patch sizes on prediction accuracy. Furthermore, we have also discussed the model's sensitivity to hyperparameters involved in entropy computation, clarifying how variations in the key parameters influence the model's overall forecasting stability.


5. We have incorporated detailed discussions addressing the following points:
- The definition and scope of the problem targeted in this work, especially with regards to the sparsity and irregularity of the time series data.
- We have clarified that our proposed method can handle both multiple target variables and sparse covariates.
- We have discussed the applicability of our proposed method, including its performance in scenarios when: (1) Inter-variable dependencies are weak; (2) Exogenous covariates are absent; (3) Exogenous covariates are noisy.

6. We have attached our source code in this submission, and we will make our code publicly available upon acceptance.

---

### Meta-Review · Area_Chair_9pdD · 2026-01-11

**Summary:**

This paper tackles Irregular Multivariate Time Series (IMTS) by formulating as a conditional generation problem that leverages pseudo-future exogenous covariates. Adopting entropy-aware adaptive patching. Achieve competitive results in real-world scenario task. However, the contribution is primarily architectural, and the necessity of using a complex diffusion-based model is not fully justified, while the treatment of irregularity remains limited. The proposed model is a diffusion variations of existing models. That is, dynamic time windows and covariance based prediction are well addressed in the literature.

**Reviewer Concerns:**

The main concerns of reviews are (1) novelty is limited; (2) It is unclear where the model makes use of the irregularity of the time series; and (3) the proposed model does not consistently outperform simpler models on long-term forecasting tasks. Authors provide additional experiments on time complexity and experimental comparisons. However, the issues raised are not addressed completely.

**Reviewer Scores:**

Due to the reasons above, the increase of reviews would be limited.

---

### Decision · Program_Chairs · 2026-01-26

Reject